# Prevalence of Alcohol-Related Harms in Yi and Han Ethnic Groups in a Prefecture in Yunnan Province, China

**DOI:** 10.3390/ijerph192316081

**Published:** 2022-12-01

**Authors:** Zhen Yu, Liping He, Wit Wichaidit, Jing Li, Ying Song, Sawitri Assanangkornchai

**Affiliations:** 1School of Public Health, Kunming Medical University, Kunming 650500, China; 2Department of Epidemiology, Faculty of Medicine, Prince of Songkla University, Songkhla 90110, Thailand

**Keywords:** alcohol, alcohol-related harm, harm from others, ethnicity

## Abstract

Background: Although differences in the prevalence of alcohol-related harm between ethnic minority and majority groups have been reported in many countries, such data are scarce in China. The findings of such assessment can provide empirical data to inform stakeholders in prioritization and allocation of resources for programs to manage and control alcohol-related problems. The objective of this study is to compare the prevalence of alcohol-related harm from others among Han and Yi populations in the Chuxiong Yi Autonomous Prefecture, Yunnan Province, China. Method: We conducted a cross-sectional study in 1370 households from 21 villages. Enumerators used convenient sampling to recruit one person aged 18 years or older from each selected household, obtained informed consent to participate, and conducted an interview using a structured questionnaire. The questionnaire included three parts: (1) demographic characteristics of the participant (including ethnic identity); (2) history of alcohol-related harm from other in the past 12 months, and; (3) drinking behaviors. We analyzed data using descriptive statistics and multivariate regression analyses, stratified by sex of the participant. Results: The prevalence of experiencing alcohol-related harm from others in Han men, Yi men, Han females, and Yi females, were 69.9%, 62.1%, 75.3%, and 63.4%, respectively. The Han vs. Yi disparity was higher among females (Adjusted OR = 2.06; 95% CI = 1.41, 3.01) than males (Adjusted OR = 1.47; 95% CI = 1.05, 2.07). The most common type of harm was feeling scared or threatened (36.9% among males, 32.4% among females) and the least common type was financial difficulty (3% among males, and 3.3% among females). Conclusions: Yi ethnic minorities in Yunnan Province had lower prevalence of alcohol-related harm from others than Han persons in the same region. However, measurement and translation-related issues of the study instrument and limited generalizability should be considered as caveats in the interpretation of the study findings.

## 1. Background

Alcohol consumption is a harmful behavior with both direct and indirect effects on health. Direct and indirect effects on the drinkers include socioeconomic consequences and substantial health problems, such as alcohol-related injuries, loss of productiveness, mental health problems, and interpersonal violence, all of which incur costs to the healthcare sector [1,2]. Alcohol consumption can also harm others who are in contact with the drinkers. Among those affected, this phenomenon is called alcohol-related harm from others (ARHFO), which may include property damages, alcohol-related injuries, accidents, and physical-verbal-sexual violence [1].

In China, alcohol is traditionally consumed during major social events, such as the spring festival, wedding ceremonies, and birthday parties. Access to alcohol has increased amidst recent economic growth, and Chinese people now drink alcohol in their daily life to relieve stress, facilitate social interaction, and foster workplace relations [3,4]. Prevalence of drinking in China has been growing since 2000, and annual per capita alcohol consumption rose from 4.1 L in 2005 to 7.2 L in 2016 [5]. Despite this increase, studies on ARHFO in China have focused mainly on driving under the influence of alcohol (DUI) [6,7,8]. Moreover, although existing legal codes prohibit businesses from selling alcohol to those under 18 years of age, China does not have a legal code that specifies the minimum age for drinking, weakening its efforts to control underage drinking.

Furthermore, China is a multi-ethnic country, and substantial socioeconomic and health disparities exist between the majority Han people and ethnic minority groups. As ethnic minority groups in other countries tend to have significantly different prevalence of alcohol-related harm than the majority of the population [9,10,11,12], we hypothesized that such discrepancy would also be found in China. A previous study found that Yi ethnic minority people had a more tolerant attitude towards drinking [13]. Besides, alcohol consumption, including harmful drinking, is significantly higher among Yi people than the overall level in China [14,15]. Thus, we supposed that Yi persons would have greater frequencies of alcohol-related harm than Han persons. However, there is a lack of data on inter-ethnic disparities in alcohol-related harm from others between Han vs. Yi peoples. Such information can provide empirical data for prioritization and allocation of resources for public health programs to manage and control alcohol-related problems. The objective of this study is to compare the prevalence of ARHFO among Han and Yi populations in a rural area of Chuxiong Yi Autonomous Prefecture, Yunnan Province, China.

## 2. Methods

### 2.1. Study Design, Study Participants, and Selection Criteria

We conducted a cross-sectional study in the Chuxiong Yi Autonomous Prefecture of Yunnan Province in southwestern China. Yunnan Province is one of the regions with the highest ethnic diversity and retention of ethnic cultures. The Prefecture has a population of 2.7 million, 72% of whom live in rural areas, 64% are Han Chinese, and 29% are Yi ethnic minority.

Our study participants included Yi and Han people who resided in Yi or Han villages in Chuxiong Yi. Inclusion criteria were (1) having resided in the village for at least 5 years; (2) aged 18 years or above, and; (3) being of Yi or Han ethnicity. Exclusion criteria were: (1) being unable to communicate with the investigators (e.g., being deaf or mute), or; (2) being in a state of mind that did not allow for clear thoughts or expression of one’s own opinions.

### 2.2. Study Variables and Measurements

(1)Outcome (Dependent Variable): Alcohol-Related Harm from Others’ Drinking

We measured frequency of self-reported harm from others’ drinking based on 10 questions (Table 1), which we adapted from an English-language questionnaire used in a WHO/Thai Health International Collaborative Research Project [16]. Each question on experience of harm had three possible choices: (0) Never experienced such harm in the past 12 months; (1) Experienced such harm 1–2 times, and; (2) Experienced such harm 3 times or more. We considered participants who reported experiencing at least one of the ten types of harm in the past year as those who experienced alcohol-related harm from others. The questions used in the mentioned Project were validated in the Project’s study setting, thus the author decided to adapt them for the study and translated them to the Chinese language with pilot-testing before launch.

(2)Exposure (Independent Variable): Ethnicity

In our survey, Yi village was defined as a village where more than 60% of the residents were of Yi ethnicity, otherwise a village would be defined as a Han village. Yi and Han households were randomly selected from Yi and Han villages. However, because inter-ethnic marriages occurred in the study area, we decided to identify a household as being Yi or Han households according to the ethnicity of the household head. Participant’s ethnicity identity was also based on ethnic identity of the household head.

(3)Covariates

Our study questionnaire also included characteristics of the study participants, including gender, age, education, residential location, and alcohol use disorder level. Most measurement questions were based on commonly-used questions in previously conducted studies in the region. We measured alcohol use disorder based on the score of the Alcohol Use Disorders Identification Test (AUDIT) as follows: (1) Non-drinking or low-risk drinking (AUDIT score of 0 to 7); (2) Hazardous or harmful drinking (AUDIT score of 8 to 14); and (3) Probable alcohol dependence (AUDIT score of 15 to 40) [17].

### 2.3. Questionnaire Development

The principal investigator and 2 team members with good command of English translated the outcome measurement questions from English to Chinese and developed the rest of the questionnaire in Chinese. The investigators then pilot-tested the questionnaire with 5 local staff and 10 local residents, and used the feedback from the pilot study participants to further modify the wording of the questionnaire to improve comprehensibility. Investigators also modified answer choices to better suit the local contexts (adding common types of local liquor, drinking places, etc.).

### 2.4. Sampling Methodology

We used multi-stage stratified random sampling to sample 1370 participants from 21 Yi and Han villages. In the first stage, investigators identified a village as either “Yi” or “Han” village from the government’s list of villages, and randomly selected 7 Yi villages and 14 Han villages. In the second stage, investigators requested the list of residents from selected villages, and used systematic random sampling to select 80 households from each Yi village and 60 households from each Han village according to the village’s list of residents.

### 2.5. Data Collection

Investigators trained 12 students from the Faculty of Public Health of Kunming Medical University as field enumerators. The training session lasted 3 days and included field investigation skills and the pilot-test of the study questionnaire.

Investigators contacted the administrative committee of each selected village, and asked village committee members to contact the heads of the selected households for permission to visit their home at least 1 day before data collection. On the day of data collection, investigators grouped the 12 field enumerators into 6 teams of 2 members. Each data collection team was escorted by one village committee member to enter the selected households. The enumerators requested permission to talk to an adult in the household who met the study criteria, then informed the potential participant about the survey and asked for the potential participant’s interest in participation. If the potential participants agreed, they would give informed consent, and the enumerators then conducted a face-to-face interview. If the selected household member refused or was not able to answer the interview, the enumerators would select another family member or a next-door neighbor who also met the study criteria to replace him or her. After the interview, the participant received 30 CNY as compensation.

### 2.6. Data Management and Data Analyses

Four investigators performed double entry of the data in the completed study questionnaires using EpiData 3.1. The investigators removed duplicated and incomplete entries during this process, then analyzed the study data using SPSS 25.0 [license number: D0EJNLL]. Descriptive statistics was used to assess differences in demographic and drinking status between Han and Yi participants. Multivariate logistic regression was used to assess the difference between Han and Yi participants with regards to ARHFO.

As age, education, socioeconomic status, and level of alcohol use disorder have all been identified as predictors of ARHFO in previous studies [18,19], we included these participant characteristics as potential confounders in multivariate logistic regression. Furthermore, because the social norm of drinking and experience of ARHFO differ widely between males and females [20], we stratified all assessments of the association between ethnicity and ARHFO by sex.

### 2.7. Ethical Considerations

The investigators did not instruct the enumerators to probe for details of the harm or the perpetrator due to the sensitivity of the study issue. However, the enumerators provided information about authorities who could provide assistance to all participants who reported experiencing any type of alcohol-related harm from others. This study received ethical approval from the Human Research Ethics Committee of Prince of Songkla University (REC 61-366-18-1).

## 3. Results

### 3.1. Characteristics of the Participants

Of 1370 participants, 793 were Han and 577 were Yi (Table 2). Han participants were more likely than Yi participants to have finished secondary school and tertiary education, whereas Yi participants were more likely than Han participants to live in a mountainous area. Han households also had a slightly higher but significant proportion of older adults and the elderly.

### 3.2. The Harms from Others’ Drinking in Yi and Han People

Han males were more likely to report experience of alcohol-related harm from others than Yi males (69.9% vs. 62.1%, respectively, *p*-value = 0.03) (Table 3). The common types of harm experienced by males included feeling scared or threatened, serious quarrel, and traffic accidents. However, the differences in each type of harm were all not statistically significant.

Similarly, Han females were more likely to report experience of alcohol-related harm from others than Yi females (75.3% vs. 63.4%, respectively, *p*-value = 0.001) (Table 4). The most common types of harm experienced by females were also feeling scared or threatened, serious quarrel, and traffic accident. However, the prevalence of feeling scared or threatened were higher among females than among males of the same ethnicity, and the prevalence of serious quarrel and traffic accident was lower among females than among males of the same ethnicity. Furthermore, the differences between Han and Yi females with regards to serious quarrel and being pushed or shoved were both statistically significant.

### 3.3. Association between Ethnicity (Yi vs. Han) and Alcohol-Related Harms

Bivariate and multivariate logistic regression analyses showed that Han males were significantly more likely than Yi males to report past-year alcohol-related harm from others, and the association remained statistically significant after adjusting for potential confounders (Adjusted OR = 1.47; 95% CI = 1.05, 2.07) (Table 5). Similarly, Han females were significantly more likely than Yi females to report past-year alcohol-related harm from others (Adjusted OR = 2.06; 95% CI = 1.41, 3.01). The difference between Han vs. Yi females was greater than the difference between Han vs. Yi men.

## 4. Discussion

We tested our hypothesis that males and females of the Yi ethnic minority group in a prefecture in Yunnan Province, China, were more likely to experience ARHFO than males and females of the majority Han ethnicity in the same region. Our study findings refuted our hypothesis: Han males and females were more likely to report past-year ARHFO than their Yi counterparts. However, a number of issues should be considered to contextualize the study findings and interpret the findings in a more thorough and careful manner.

The prevalence of ARHFO in our study was relatively high compared to other countries in Southeast Asia, including Thailand, Sri Lanka, India, Vietnam, and Lao PDR [21]. However, our measurement question on experience of serious quarrel was subjective, as it put the burden of subjectively defining “serious quarrel” on the participants. Similarly, response to the question on feeling unsafe was open to how each participant defined being “threatened or scared”, and that the behaviors were of the “drinkers”. In other words, the definition of fear, and the definition of whether the offending person was under the influence of alcohol, depended on each participant. Yi ethnic minority persons who practiced traditional Bimoism religion might have different narratives of suffering and fear compared to their Han counterparts [22]. Bimoist Yi tend to commonly attribute bad things to external interventions, and thus might have regarded the ARHFO that they experienced as an act of divine intervention rather than an act by the drinkers themselves, and might have under-reported their experience of harm from others, thus biasing our measure of association away from the null value. Future studies should consider modifying the measurement questions to be more objective and consider adjusting for the religiosity and beliefs of the study participants.

The differences between the sexes in our study suggested that sex was a potential effect modifier in the association between ethnicity and alcohol-related harm from others. The role of sex in ARHFO has been reported in many previous studies, which found that males were more likely to be involved in violence than females [23,24,25]. Chinese males also have more time to socialize than Chinese females [26], and drinking is the most common way to socialize. Alcohol-related expectations could have been equally present among Han males and Yi men, making the gap narrower than that between Han females and Yi females. Overall, Han females were the most likely to have reported ARHFO. This high prevalence might have been influenced by environmental factors, such as community crowding, street drinking, or drinking in public places in Han villages [27]. Han villages are mainly located in the flat area close to the city while 50% of Yi villages are in the remote mountain area. Furthermore, Yi society traditionally consists of patrilineal clans [28], thus Yi females may have less need to meet strangers and socialize than Han females, which also reduced the risk of ARHFO. The ability to disclose their fear and sufferings to interviewers who are unfamiliar people outside the family or village might be another factor determining differences in reporting harm between ethnic groups and sexes. Han people, especially females who were more exposed to the outside world, might have felt more comfortable to disclose incidents in their family to strangers, influencing the level of alcohol-related harm from others.

Although there was no statistically significant difference between Han and Yi peoples, the prevalence of harm from alcohol-related traffic accidents in the past 12 months was concerning. Drinking-and-driving is generally attributed to complacency or misperception of one’s driving ability after alcohol consumption [29,30,31]. Risk behaviors after alcohol consumption are influenced by cultural, psychological, and biological factors. Assessment of existing measures to control drinking-and-driving, as well as further studies with a combination of quantitative and qualitative methods, may be needed in order to understand the context-specific drivers of alcohol-related harm in this region and design interventions that can yield much-needed reduction in alcohol-related harm from traffic accidents.

Our study provides new information on alcohol-related harm in a prefecture of Yunnan Province and a basis for continuing exploration of the hazards of ARHFO. However, a number of limitations in our study should be considered in the interpretation of its findings. First of all, our measurement of past-year experiences could have been affected by the participants’ memory, perception, and definition of harm [32]. Secondly, the questionnaire used in our survey was translated from the WHO-Thai Alcohol-Related Harm Research Project questionnaire [16], and some questions might have slightly differed when translated from English into Mandarin Chinese and Yi dialects. These differences could have affected the interpretation of the questions and introduced a moderate level of information bias into the study findings. Thirdly, our study was conducted only in a prefecture in Yunnan Province, which limited the generalizability to other settings or to a broader (provincial/national/global) level.

## 5. Conclusions

Contrary to our hypothesis, the Yi ethnic minority in Yunnan Province had lower prevalence of ARHFO than Han persons in the same region. This difference was greater among Han vs. Yi females than among Han vs. Yi men. However, issues pertaining to measurement and translation of the study instruments, as well as limited generalizability, should be considered as caveats in the interpretation of the study findings. 

## Figures and Tables

**Table 1 ijerph-19-16081-t001:** Type of harms occurring to the participants from other’s drinking.

Question: Did you experience the following problems in the last 12 months because of other people’s drinking?	
**Type of Harm**	**Description**
1. Serious quarrel	Having serious quarrel with any drinker (without physical harm)
2. Being pushed or shoved	Being pushed or shoved by any drinker (or slight body hit)
3. Severe physical harm	Having severe physical harm from any drinker (bleeding, large subcutaneous bleeding, and severe pain that have to see doctor)
4. Property loss	Losing any property, including property being broken or lost
5. Traffic accident	Suffering a traffic accident by a drunk-driver
6. Feeling scared	Feeling threatened or scared by other drinker’s behavior, such as being shouted at, threatening remarks, or aggressive gestures being made
7. Being harassed	Being continuously engaged in an unwelcomed conversation despite refusal, being mocked, or experiencing other indecent behaviors from an intoxicated person
8. Unpleasant incidents	Having unpleasant incidents with friends or neighbors
9. Being pushed to drink	Being pushed to drink at a party
10. Financial problem	Having financial difficulty in family, with negative impact on normal life, family planning, and/or personal planning

**Table 2 ijerph-19-16081-t002:** Participant socio-demographic characteristics and drinking status, in frequencies and percentages.

Variable	Han (N = 793)	Yi (N = 577)	*p*-Value
Sex			0.243
Male	428 (54.0)	293 (50.8)	
Female	365 (46.0)	284 (49.2)	
Education			<0.001
Primary school or illiterate	292 (36.8)	294 (51.0)	
Secondary school	381 (48.0)	222 (38.5)	
College or above	120 (15.1)	61 (10.6)	
Age			0.001
Young adult (<39 years)	234 (29.5)	212 (36.8)	
Older adult (40–59 years)	429 (54.1)	294 (51.0)	
Elderly (≥60 years)	130 (16.4)	70 (12.2)	
missing	0 (0)	1 (0.2)	
Residential location			<0.001
Mountainous area	72 (9.1)	194 (33.6)	
Plateau area	721 (90.9)	383 (66.4)	
Drink status			0.698
Non or low-risk drinking	495 (62.4)	363 (62.9)	
Hazardous-harmful drinking	91 (11.5)	74 (12.8)	
Probable dependent drinking	207 (26.1)	140 (24.3)	

**Table 3 ijerph-19-16081-t003:** Frequency of harm from other’s drinking in the past 12 months among Han and Yi males in rural areas of Chuxiong Yi Autonomous Prefecture (n = 721).

Type of Harm	Han Male (N = 428)			Yi Male (N = 293)			*p*-Value
	None	1–2 Times	>3 Times	None	1–2 Times	>3 Times	
1. Serious quarrel	337 (78.7)	80 (18.7)	11 (2.6)	241 (82.3)	48 (16.4)	4 (1.4)	0.225
2. Being pushed or shoved	385 (90.0)	36 (8.4)	7 (1.6)	262 (89.4)	28 (9.6)	3 (1.0)	0.841
3. Severe physical harm	407 (95.1)	21 (4.9)	0 (0)	282 (96.2)	8 (2.7)	3 (1.0)	0.481
4. Losing any property	393 (91.8)	33 (7.7)	2 (0.5)	277 (28.3)	13 (4.4)	3 (1.0)	0.172
5. Traffic accident	361 (84.3)	58 (13.6)	9 (2.1)	256 (87.4)	27 (9.2)	10 (3.4)	0.302
6. Feeling scared or threatened	270 (63.1)	117 (27.3)	41 (9.6)	198 (67.6)	69 (23.5)	26 (8.9)	0.983
7. Being harassed	381 (89.0)	41 (9.6)	6 (1.4)	261 (89.1)	25 (8.5)	7 (2.4)	0.241
8. Problems with friends or neighbors	397 (92.8)	28 (6.5)	3 (0.7)	275 (93.9)	14 (4.8)	4 (1.4)	0.585
9. Being pushed to drink	398 (93.0)	23 (5.4)	7 (1.6)	278 (94.8)	13 (4.4)	2 (0.7)	0.294
10. Financial difficulty	415 (97.0)	12 (2.8)	1 (0.2)	283 (96.6)	7 (2.3)	3 (1.0)	0.765
At least one of all the above harms	122 (30.1)	306 (69.9)		105 (37.9)	188 (62.1)		0.030

**Table 4 ijerph-19-16081-t004:** Frequency of harm from other’s drinking in the past 12 months among Han and Yi females in rural areas of Chuxiong Yi Autonomous Prefecture (n = 649).

Type of Harm	Han Female (N = 365)			Yi Female (N = 284)			*p*-Value
	None	1–2 Times	>3 Times	None	1–2 Times	>3 Times	
1. Serious quarrel	296 (81.1)	57 (15.6)	12 (3.3)	249 (87.7)	25 (8.8)	10 (3.5)	0.03
2. Being pushed or shoved ^△^	335 (91.8)	25 (6.8)	5 (1.4)	272 (95.8)	10 (3.5)	2 (0.7)	0.04
3. Severe physical harm	347 (95.1)	16 (4.4)	2 (0.5)	278 (97.9)	5 (1.8)	1 (0.4)	0.06
4. Losing any property	344 (94.2)	20 (5.5)	1 (0.3)	274 (96.5)	10 (3.5)	0 (0)	0.184
5. Traffic accident * ^△^	325 (89.0)	38 (10.4)	2 (0.5)	262 (92.3)	19 (6.7)	3 (1.1)	0.178
6. Feeling scared or threatened *^△^	170 (46.6)	148 (40.5)	47 (12.9)	160 (56.3)	89 (31.3)	35 (12.3)	0.961
7. Being harassed * ^△^	344 (94.2)	19 (5.2)	2 (0.5)	268 (94.4)	13 (4.6)	3 (1.1)	0.033
8. Problems with friends or neighbors	350 (95.9)	11 (3.0)	4 (1.1)	271 (95.4)	10 (3.5)	3 (1.1)	0.774
9. Being pushed to drink	349 (95.6)	14 (3.8)	2 (0.5)	277 (97.5)	3 (1.1)	4 (1.4)	0.201
10. Financial difficulty	354 (97.0)	6 (1.6)	5 (1.4)	278 (97.9)	6 (2.1)	0 (0)	0.462
At least one of all the above harms	90 (24.7)	275 (75.3)		103 (36.6)	181 (63.4)		0.001

* *p* value < 0.05, Han women compared with the data of Han men. ^△^
*p* value < 0.05, Yi women compared with the data of Yi men.

**Table 5 ijerph-19-16081-t005:** Associations between ethnicity and frequency of harm from other’s drinking with adjustment for other demographic and socioeconomic characteristics, stratified by sex.

Characteristic	Frequency of ARHFO (*n* (%))		Crude OR (95%CI)	Adjusted OR (95%CI) *
	Never	Ever		
Among males (n = 721)				
Yi	111 (37.9)	182 (62.1)	Ref.	Ref.
Han	129 (30.1)	299 (69.9)	1.41 (1.03, 1.93)	1.47 (1.05, 2.07)
Among females (n = 649)				
Yi	104 (36.6)	180 (63.4)	Ref.	Ref.
Han	90 (24.7)	275 (75.3)	1.77 (1.26, 2.48)	2.06 (1.41, 3.01)

* From binary logistic regression with adjustment for age, education, residential location, and drinking status, ARHFO = alcohol-related harm from others, OR = odds ratio.

## Data Availability

De-identified and anonymized data presented in this study are available through reasonable request to the corresponding author.

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
