# Peer review of "Prevalence of Alcohol-Related Harms in Yi and Han Ethnic Groups in a Prefecture in Yunnan Province, China"

_ijerph, 2022, doi:10.3390/ijerph192316081_

Round 1
Reviewer 1 Report
Here, the authors describe a cross-sectional study of one member from each of 1,370 households in 21 villages in Yunnan Province, China, assessing the frequencies of alcohol-related harms in participants in two ethnic groups aged 18 or older. There were high frequencies of alcohol-related harms in both ethnic groups, but higher among the Han than Yi participants, particularly Han females. The authors note that measurement and translation issues may be problematic for the study.
Please specify the nature of the alcohol-related harms assessed in the Abstract. In particular, it would be good to specify the most common alcohol-related harms reported by participants.
It is stated in the Discussion that the authors hypothesised the Yi would have greater frequencies of alcohol-related harm than the Han. Please include this hypothesis in the Introduction. In addition, please provide justification for why this is the expectation.
In Table 1, in the Harassed section, one of the descriptions is “being accosted after saying no”. What is this in regard to?
Please use sex and male/female terminology, rather than gender and men/women.
Reviewer 2 Report
First of all congratulations to the authors. They have carried out important field work and widely synthesized the existing state of the art
1. I would like to make two suggestions, one contextual and the other methodological as an aspect of improvementAlcohol consumption has been addressed since the age of 18. It would be interesting to clarify from what age, the state context allows the consumption of alcohol as a justification for the introduction
2. It would be interesting to include within the methodology, the validity of the questionnaires used. This fact would clarify in the reader that said questionnaire is appropriate for the measurements in our research and sample.
Thank you very much for your work, it has been easy to read and with very significant results in the face of this important public health problem.
